# Learning Numerical Attributes in Knowledge Bases

**Bhushan Kotnis**                                                       BHUSHAN.KOTNIS@NECLAB.EU
**Alberto García-Durán**                                                  ALBERTO.DURAN@NECLAB.EU
*NEC Labs Europe*
*Kurfürsten-Anlage 36, 69115 Heidelberg, Germany*

## Abstract

Knowledge bases (KB) are often represented as a collection of facts in the form (HEAD, PREDICATE, TAIL), where HEAD and TAIL are entities while PREDICATE is a binary relationship that links the two. It is a well-known fact that knowledge bases are far from complete, and hence the plethora of research on KB completion methods, specifically on link prediction. However, though frequently ignored, these repositories also contain numerical facts. Numerical facts link entities to numerical values via numerical predicates; e.g., (PARIS, LATITUDE, 48.8). Likewise, numerical facts also suffer from the incompleteness problem. To address this issue, we introduce the numerical attribute prediction problem. This problem involves a new type of query where the relationship is a numerical predicate. Consequently, and contrary to link prediction, the answer to this query is a numerical value. We argue that the numerical values associated with entities explain, to some extent, the relational structure of the knowledge base. Therefore, we leverage knowledge base embedding methods to learn representations that are useful predictors for the numerical attributes. An extensive set of experiments on benchmark versions of FREEBASE and YAGO show that our approaches largely outperform sensible baselines. We make the datasets available under a permissive BSD-3 license.

## 1. Introduction

Knowledge Bases (KBs) are playing an increasingly important role in a number of AI applications. KBs can be seen as a collection of facts or triples of the form (HEAD, PREDICATE, TAIL), denoted as $(\mathtt{h}, \mathtt{p}, \mathtt{t})$, where HEAD and TAIL correspond to entities and PREDICATE corresponds to a relationship that holds between these two entities. This structured information is easily accessible by AI systems to enhance their performance. A variety of AI applications such as recommender systems, natural language chatbots or question answering models, have benefited from the rich structural information archived in these repositories. This is because much of human knowledge can be expressed with one or more conjunctions of knowledge facts.

However, KBs' capabilities are limited due to their incompleteness[1]. Consequently there has been a flurry of research on knowledge base completion methods in recent years. Relationship extraction [Riedel et al., 2013] (i.e., classification of semantic relationship mentions), knowledge graph matching [Suchanek et al., 2011, Lacoste-Julien et al., 2013] (i.e., alignment and integration of entities and predicates across KBs), or search-based question-

---

1. Freebase, which is likely the most popular knowledge base, illustrates this problem. For example, the relation type /PERSON/NATIONALITY is not present in around 78% of the entities representing people [Min et al., 2013]. Similarly, it contains only around 40% of the completions to the query (USA, /LOCATION/CONTAINS, ?) [Garcia-Duran and Niepert, 2018].

answering [West et al., 2014] (i.e., queries issued to a web search engine) are a few different ways to address the incompleteness problem. However, the literature on the so-called link prediction methods [Nickel et al., 2016] has received more attention in the last few years in comparison to the aforementioned approaches. Contrary to other solutions, link prediction methods aim to find missing links between entities exclusively based on the existing information contained in the KB. This is achieved by ranking entities that are answer candidates for the query. The queries these methods typically address are of the form (USA, /LOCATION/CONTAINS, ?), or (MADRID, /LOCATION/CAPITALOF, ?), whereas the missing element –represented by a question mark– is an entity contained in the KB.

Many link prediction methods only harness feature types learned from the rich relational information contained in the KB to infer new links, and only very recently [Garcia-Duran and Niepert, 2018, Pezeshkpour et al., 2018] numerical attributes have been integrated along with other feature types to improve link prediction performance. Similarly, numerical information is also represented as facts such as (BERLIN, /LOCATION/LATITUDE, 52.31) or (ALBERT_EINSTEIN, /PERSON/BIRTH_YEAR, 1879). However, as shown in [Garcia-Duran and Niepert, 2018] the application of numerical attributes is limited because of the same incompleteness problem: Many entities are missing numerical attribute values they are expected to possess. For example, entities that represent locations should have numerical information regarding latitude, longitude or area, among others; whereas for entities representing people, numerical predicates such as the birth year, weight or height would be more appropriate. Figure 1 illustrates an example of a KB where some entities have missing numerical attributes.

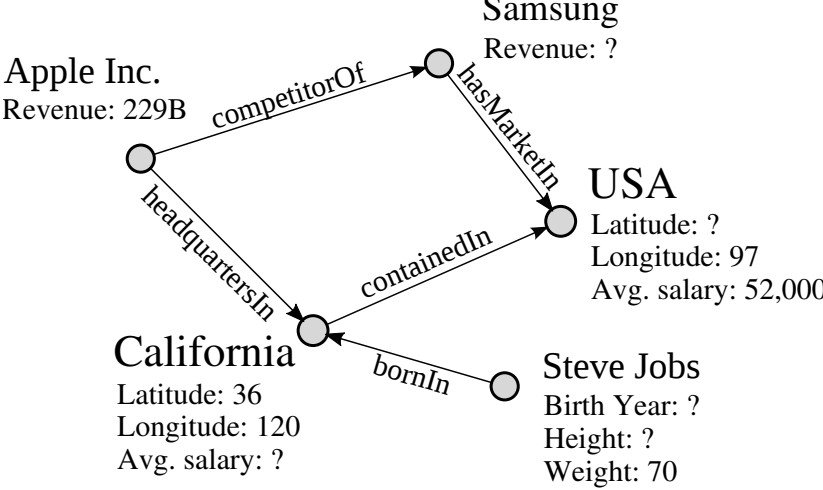

Figure 1: A small part of a knowledge base. Some entities have missing numerical predicates. For the sake of visualization numerical predicates are represented as entities' attributes, instead of as links.

In this work we focus on the problem of completing queries where the relationship is a numerical predicate. Consequently, the answer to this new type of query is a numerical value. This is contrary to the link prediction problem, wherein the answer to a query is

always an element of a closed vocabulary. Examples of queries addressed in this paper are (Apple_Inc., revenue, ?) or (California, average_salary, ?). While one can interpret link prediction as a classification/ranking problem, this is rather a regression problem.

The main contributions of this paper are:

- We introduce the problem of predicting the value of entities' numerical attributes in KBs. For the sake of simplicity we term this as 'numerical attribute prediction problem'. To our knowledge, this is the first time this problem is addressed in the literature.

- We create benchmark datasets for this problem. We use well-known subsets of Freebase and Yago as the blueprints for creating these benchmarks. We also create versions of these datasets for different percentages of sparsity by artificially removing facts that involve numerical predicates. All these benchmark datasets will be made publicly available.

- We propose two meaningful baselines for this problem. These baselines are inspired by previous work done in the node classification and the imputation literature.

- We propose supervised and semi-supervised approaches to this problem. The semi-supervised approaches significantly outperform the baselines in all datasets and conditions.

The paper is organized as follows: We discuss the related work in Section 2. Afterwards we formalize the problem of predicting numerical values for entities' numerical attributes in KBs in Section 3. We describe our approaches to this problem, as well as the two baselines. Section 4 reports the experimental setting followed by an extensive set of experiments on different datasets with different degrees of sparsity in Section 5. Finally, we summarize the conclusions of our study in Section 6.

## 2. Related Work

There is an extensive body of work on link prediction [Bordes et al., 2013, Yang et al., 2014, Nickel et al., 2016, Trouillon et al., 2017]. Logical approaches [Russell and Norvig, 2016] operate on a set of logical rules that are usually handcrafted and/or mined. These logical formulas are evaluated between entity pairs to generate feature representations which are then used for a downstream machine learning model. On the other hand, KB embedding methods [Nickel et al., 2016] learn feature representations –embeddings– for all elements in a KG by optimizing a designed scoring function. Given a fact, these scoring functions output a score that relates to the likelihood of that fact being true. A popular and successful instance of KB embedding method is TransE [Bordes et al., 2013], where predicates are modeled as translations in the entity embedding space.

Much less work has been done on entity-type classification [Moon et al., 2017, Yogatama et al., 2015]. This problem is inherently related to link prediction, since it amounts to complete queries of the form (head, typeOf, ?), where the question mark corresponds to a certain entity type (e.g. location, artist, ...). Therefore, link prediction and entity-type

classification share certain similarities with the numerical attribute prediction problem. Most importantly, they all make use of the relational information for KB completion, one way or another. However, there is a crucial difference between link prediction and numerical attribute prediction. In the former, a query can be completed with one or several elements contained in a relatively small vocabulary, whereas in the later the answer may (potentially) take an infinite number of real values.

There is another line of research related to our work, namely value imputation [Rubin, 1976]. In statistics, imputation is the process of replacing missing data with substituted values. In the simplest case, one can replace the missing values of a variable by the mean of all existing values of that variable. This technique is called mean imputation. It preserves the mean of the variable, but alters the underlying variable distribution to be more peaked at the mean [Allison, 1999]. However, it is the most commonly practiced approach for value imputation [Rubin et al., 2007], and it has been shown to be competitive for a number of downstream tasks [Beaulieu-Jones and Moore, 2017, Malone et al., 2018]. Another popular approach is called regression imputation, where the missing values of a variable are estimated by a regression model from the observed values of other variables.

There is some work on using text for predicting numerical attributes of entities such as [Davidov and Rappoport, 2010, Gupta et al., 2015]. [Gupta et al., 2015] uses Word2Vec embeddings of named entities as inputs to a number of regression models. Similar to us, they aim to predict numerical attributes of knowledge base entities. Different to us, they leverage text information to do so. This difference is important, because we do not assume the existence of information other than the graph structure. Our problem is general enough to address knowledge bases where entities names are unknown or anonymized (e.g. medical knowledge bases).

To our knowledge there is no existing work in the value imputation literature that attempts to fill missing values in KBs while taking advantage of the structural information provided by the KB.

## 3. Prediction of Numerical Attributes

A knowledge base, KB, is denoted as $\mathcal{G} = (\mathcal{E}, \mathcal{P})$, where $\mathcal{E}$ is a set of entities and, $\mathcal{P}$ is a set of relation types or predicates. This standard definition can be found in many papers in the link prediction literature [Nickel et al., 2016, Garcia-Duran and Niepert, 2018]. A KB is a collection of **facts** (or **standard facts**) $(\mathtt{h}, \mathtt{p}, \mathtt{t})$ where $\mathtt{p} \in \mathcal{P}$ and $\mathtt{h}, \mathtt{t} \in \mathcal{E}$. We now define a knowledge base enriched with numerical attributes as $\mathcal{G}_{NA} = (\mathcal{G}, \mathcal{A}, \mathcal{N})$. Entities in $\mathcal{G}$ are associated with numerical values $\mathcal{N}$ via numerical predicates $\mathcal{A}$. This information can be expressed as a collection of **numerical facts** $(\mathtt{h}, \mathtt{a}, \mathtt{t})$ where $\mathtt{a} \in \mathcal{A}$, $\mathtt{h} \in \mathcal{E}$ and $\mathtt{t} \in \mathcal{N}$. In the paper we interchangeably use the term 'numerical predicate' with 'numerical attribute'. The numerical attribute prediction problem seeks the most probable completion of a fact $(\mathtt{h}, \mathtt{a}, ?)$, where $\mathtt{h} \in \mathcal{E}$, $\mathtt{a} \in \mathcal{A}$ and $? \in \mathcal{N}$.

We refer to the set of entities for which the value of the numerical attribute $a$ is known as $\mathcal{E}^a \subseteq \mathcal{E}$. Let $e$ be an entity with numerical attribute $a$, then we denote the known numerical value for attribute $a$ as $\mathtt{n}_e^a$. The goal is to learn a function $f : \mathcal{E} \to \Re$, $\Re$ denotes the set of reals.

One can omit the relational information given by the graph $\mathcal{G}$ and apply a value imputation method to fill missing values. However, it is intuitive to assume the existence of an underlying generative model that (partially) determines the relational structure of the KB based on the values of the entities' numerical attributes. For instance, two entities are likely linked via the relationship /LOCATION/CONTAINS if they have similar latitude and longitude; or two highly connected entities that correspond to people are likely to have similar birth years. If this assumption is true, then a model that exploits the graph structure information is likely to outperform simple value imputation methods. Nevertheless, while this may be true for a number of numerical attributes, for others the graph structure may introduce noise or, in the best case, be irrelevant.

## 3.1 Baselines

Inspired by previous work in the value imputation and the node classification literature, we propose the following baselines.

### 3.1.1 GLOBAL

A simple and natural baseline is simply using the sample mean of the attribute specific training data as a predictor for missing values. This is known as mean imputation [Rubin et al., 2007]. At test time, given an entity $e$ for which we aim to predict the value of numerical predicate $a$, denoted as $\hat{\mathfrak{n}}_e^a$, this baseline simply assigns the sample mean of all known entities possessing the same numerical attribute ($\mathcal{E}^a$). This is formally described below.

$$\hat{\mathfrak{n}}_e^a = f(\{n_{e'}^a \mid e' \in \mathcal{E}^a\}), \tag{1}$$

where $f$ is the sample mean.

We term this model as GLOBAL because it harnesses global information from the entire attribute specific training set.

In this work we use the root mean square error (RMSE) and the mean absolute error (MAE) as evaluation metrics. While the sample mean is the best estimator for the former, the sample median is the optimum for the latter [Murphy, 2012]. Consequently, in the experimental section we use median imputation when reporting the MAE and mean imputation when reporting on RMSE metrics. Median imputation is obtained by simply replacing the sample average by the median in Eq. (1).

### 3.1.2 LOCAL

Our second baseline takes into account that entities are interconnected through a relational graph structure. Thus it is natural to define a baseline that exploits the neighborhood or local graph structure.

The weighted-vote relational neighbor [Macskassy and Provost, 2003] is a relational classifier often used as a benchmark in the node classification literature. It estimates the class label of a node as a weighted average of its neighbors' class labels. Despite its simplicity, it is shown to be competitive [Perozzi et al., 2014a] and is advocated as a sensible relational classification baseline [Macskassy and Provost, 2007].

Inspired by such work, we propose an adaptation for our setting and problem. For a numerical attribute $a$, this baseline estimates a value for the entity $e$ as the average of its neighbors' attribute values for that numerical attribute. Here, the neighborhood of a node $e$, denoted $\mathcal{N}_e$, is defined as the set of nodes that are connected to $e$ through any relation type. The baseline is formalized as follows

$$\hat{\mathtt{n}}_e^a = f(\{n_{e'}^a \mid e' \in \mathcal{E}^a \cap \mathcal{N}_e) \tag{2}$$

where, as before, $f$ is either the sample mean or the sample median depending on the evaluation metric reported. We term this model as Local because it uses the local neighborhood information for prediction.

In the case where $\mathcal{E}^a \cap \mathcal{N}_e = \varnothing$, we make use of the so-called Global baseline to make a prediction.

## 3.2 Our Approaches

We leverage KB embedding methods to learn feature representations –embeddings– of entities that (ideally) are predictive for the different numerical attributes. As we argued before, this is only true if the entities' numerical attributes determine, to a certain extent, the existence of a certain relation type between two entities. We first learn knowledge base embeddings, and in a second step we use these embeddings, along with the numerical facts, to train downstream machine learning models for predicting numerical attributes. This pipeline is reminiscent of recent work [Perozzi et al., 2014b] in the node classification literature.

While there is an extent literature on KG embedding methods [Nickel et al., 2011, Bordes et al., 2013, Nickel et al., 2016], recent work [Kadlec et al., 2017] shows that well-tuned "simple" scoring functions [Yang et al., 2014] are very hard to beat. Likewise [García-Durán et al., 2016] shows that TransE [Bordes et al., 2013] performs similarly or even better than many of its variants, such as TransH [Wang et al., 2014] or TransR [Lin et al., 2015].

Due to its simplicity and good performance in related problems, we choose TransE to illustrate the generic principles behind our models. Note, however, that the methodology described is agnostic to the chosen KG embedding method.

The probability for a fact $\mathtt{d} = (\mathtt{h}, \mathtt{p}, \mathtt{t})$ being true is $p(\mathtt{d} \mid \theta) = \frac{g(\mathtt{d}|\theta)}{\sum_{\mathtt{c}} g(\mathtt{c}|\theta))}$, where $\mathtt{c}$ indexes all possible triples, and $\theta$ all learnable parameters of TransE, whose scoring function $g$ is $g(\mathtt{d} \mid \theta) = ||\mathbf{h} + \mathbf{p} - \mathbf{t}||_2$. We use bold letters $\mathbf{h}, \mathbf{p}, \mathbf{t} \in \Re^d$ to denote the corresponding $d$-dimensional feature representations of $\mathtt{h}, \mathtt{p}, \mathtt{t}$, respectively. Note that this formulation is impractical because the cost of computing all possible triples is unfeasible. Instead, for each triple $\mathtt{d} = (\mathtt{h}, \mathtt{p}, \mathtt{t}) \in \mathcal{G}$ we generate a set of $N$ triples $(\mathtt{h}, \mathtt{p}, \mathtt{t}')$ by sampling $N$ entities $\mathtt{t}'$ uniformly at random from the set of all entities. This process, which is termed as negative sampling, is repeated for the head of the triple.

For a given set of facts $\mathbf{D}$ that are part of the KB $\mathcal{G}$, the logarithmic loss is defined as

$$\mathcal{L}_\mathcal{G} = -\sum_{\mathtt{d} \in \mathbf{D}} \log p(\mathtt{d} \mid \theta). \tag{3}$$

All parameters $\theta$ are learned for minimizing $\mathcal{L}_\mathcal{G}$ with stochastic gradient descent.

Once the representation learning phase is finished we evaluate two different approaches that utilize these embeddings for addressing the numerical attribute prediction problem.

### 3.2.1 REGRESSION MODEL

In the simplest case, for each numerical attribute we use the learned feature representations as input to a regression model to predict the corresponding numerical attribute.

For numerical attribute $a$ the loss function is given by

$$\mathcal{L}_{\mathcal{R}}^a = \sum_{\mathbf{e} \in \mathcal{E}^a} (f_{\vartheta^a}(e) - \mathtt{n}_e^a)^2 + \lambda_a ||\vartheta^a||_2^2, \tag{4}$$

where $f_{\vartheta^a}$ refers to the regression function for numerical attribute $a$, $\vartheta^a$ refers to the learnable parameters of $f_{\vartheta^a}$, and $\lambda_a$ is the regularization hyper-parameter. In this work we use a linear regression model: $f_{\vartheta^a}(e) = \mathbf{e}^T \mathbf{w}^a + b^a$, where $\mathbf{w}^a \in \Re^d$ is the weight vector and $b^a$ is the corresponding bias term. At test time, given a query related to a certain numerical attribute $a$ and a certain entity $e$, the prediction is computed by applying the corresponding linear regression model: $\hat{\mathtt{n}}_e^a = f_{\vartheta^a}(e)$. We refer to this approach as LR.

### 3.2.2 NUMERICAL ATTRIBUTE PROPAGATION

Previously we defined $\mathcal{E}^a$ as the set of entities with known numerical attribute $a$. Similarly, we define $\mathcal{Q}^a$ as the set of entities with missing values for numerical attribute $a$.

We consider numerical attribute values as labels, and, consequently, we can think of $\mathcal{E}^a$ and $\mathcal{Q}^a$ as the set of labeled and unlabeled nodes, respectively. Therefore, semi-supervised learning is a natural choice because it also uses unlabeled data to infer values of numerical attributes.

Label propagation (LP) [Zhu and Ghahramani, 2002, Fujiwara and Irie, 2014] has been proven to be an effective semi-supervised learning algorithm for classification problems. The key assumption of label propagation, and in general most semi-supervised learning algorithms, is similar to ours: Data points close to each other are likely to have the same label –numerical attribute values in our case.

We aim to propagate numerical attribute information across the graph using LP. For numerical attribute $a$, we use the learned representations $\{\mathbf{e}\}_{e \in \mathcal{E}^a \cup \mathcal{Q}^a}$ to induce a $k$-nearest neighbor graph ($k$NN) using euclidean distance. This graph is characterized by an adjacency matrix $\mathbf{A} \in \Re^{N \times N}$, where $N = |\mathcal{E}^a| + |\mathcal{Q}^a|$. The edge weights of the adjacency matrix represent similarities between the connected entities, which are computed according to a similarity metric $\rho$ –in this work we use a radial basis function kernel[2].

We then compute the transition matrix $\mathbf{T}$ by row-wise normalizing the matrix $\mathbf{A}$. Without loss of generality, we arrange labeled and unlabeled data so that $\mathbf{T}$ can be decomposed as

$$\mathbf{T} = \begin{bmatrix} \mathbf{T}_{\mathcal{E}^a \mathcal{E}^a} & \mathbf{T}_{\mathcal{E}^a \mathcal{Q}^a} \\ \mathbf{T}_{\mathcal{Q}^a \mathcal{Q}^a} & \mathbf{T}_{\mathcal{Q}^a \mathcal{E}^a} \end{bmatrix}. \tag{5}$$

---

2. $\rho(\mathbf{x}, \mathbf{y}) = exp(-\dfrac{||\mathbf{x} - \mathbf{y}||^2}{\sigma})$

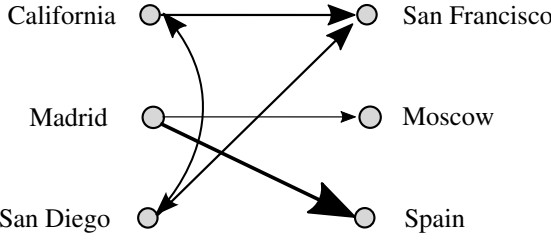

Figure 2: Illustration of transition matrix for a certain numerical attribute. Only links from unlabeled to both unlabeled and labeled entities are shown. The wider the arrow, the more likely the transition. In this example $\mathcal{E}^a = \{$San Francisco, Moscow, Spain$\}$ and $\mathcal{Q}^a = \{$California, Madrid, San Diego$\}$.

The transition matrix $\mathbf{T}$ (illustrated in Figure 2) can be iteratively used to propagate numerical information across the graph until a stopping criterion is reached. Alternatively, this problem can be solved in a closed form:

$$\hat{\mathbf{n}}^a_{\mathcal{Q}^a} = \underbrace{(\mathbf{I} + \mathbf{T}_{\mathcal{Q}^a \mathcal{Q}^a})^{-1} \mathbf{T}_{\mathcal{Q}^a \mathcal{E}^a}}_{\mathbf{M}^a} \mathbf{n}^a_{\mathcal{E}^a}, \tag{6}$$

where $\mathbf{n}^a_{\mathcal{E}^a} \in \Re^{|\mathcal{E}^a|}$ is a vector that contains all values of numerical attribute $a$ for labeled nodes. Similarly, $\hat{\mathbf{n}}^a_{\mathcal{Q}^a} \in \Re^{|\mathcal{Q}^a|}$ is a vector that contains all predicted values of numerical attribute $a$ for unlabeled nodes. We refer to the matrix $\mathbf{M}^a$ in Section 5.1. We term this as Numerical Attribute Propagation (or NAP).

Related work [Minervini et al., 2017] uses label propagation to perform link prediction in web ontologies by casting it as a binary classification problem, where the similarity graph is built based on homophilic relationships.

### 3.2.3 INJECTING NUMERICAL INFORMATION INTO THE EMBEDDINGS

In the two aforementioned solutions, we fully rely on the feature representations learned by, in this case, TRANSE to be meaningful with respect to the numerical attributes we aim to predict. This relates to our initial assumption that the relational structure of a KB can be explained, to some extent, by the numerical attributes of the entities. However, there might be cases where the values taken by entities for a certain numerical attribute do not fully relate to the relational structure of the KB.

Motivated by this consideration, we set out to answer the question: *Can these models benefit from learning feature representations incorporating, beside the graph structure, numerical attribute information?*

To answer this question we incorporate the learning objective of Eq. (4) into the learning objective of TRANSE (Eq. (3)):

$$\mathcal{L} = \mathcal{L}_{\mathcal{G}} + \alpha \sum_{a \in \mathcal{A}} \mathcal{L}^a_{\mathcal{R}}, \tag{7}$$

where $\alpha$ weights the importance of the linear regression objectives. All parameters are learned using stochastic gradient descent. We term these embeddings as TransE++, which, contrary to TransE, are also learned with numerical facts. While Nap and Lr use TransE feature representations, their counterparts Nap++ and Lr++ leverage TransE++ embeddings.

Different numerical attributes exhibit different scales of attribute values. For example 'person.height' ranges from 1 to 2 meters while 'location.area' scales from several hundred to even millions of kilometers. Bigger scales lead to larger prediction errors during training which in turn affect the back-propagation gradients used for learning the TransE++ embeddings. To alleviate this problem, we normalize each numerical attribute to zero-mean and unit-variance. We also experimented with min-max scaling, however it gave worse performance compared to standard scaling. Scaling numerical attribute values remains an interesting challenge.

For Lr++ we do not directly use the regression models learned during opimizing the TransE++'s learning objective (Eq. (7)). Instead we use the learned TransE++ embeddings to train a new regression model $\mathcal{L}_{\mathcal{R}}^a$ for each numerical attribute $a \in \mathcal{A}$. This is because of the computational difficulty in tuning hyper parameter $\lambda_a$ for each numerical attribute while learning TransE++, which we found important to obtain good performance. Note that the hyper parameter space grows exponentially with the number of attributes $|\mathcal{A}|$. For this reason, we set $\lambda_a = \lambda = 0$ in the regression objectives in Eq. (7) for learning TransE++ embeddings (first step). For the final regression models (second step) we do tune $\lambda_a$s (independently), which, though suboptimal, facilitates their tuning.

## 4. Experimental Settings

The proposed methods are evaluated by their ability to answer completion queries of the form $(\mathtt{h}, \mathtt{a}, ?)$, where $\mathtt{h} \in \mathcal{E}$, $\mathtt{a} \in \mathcal{A}$ and $? \in \mathcal{N}$. We evaluate the baselines and our models on two benchmark datasets: FB15K-237 [Toutanova et al., 2015] and YAGO15K [García-Durán et al., 2018]. While for the former, numerical attributes were introduced in [Garcia-Duran and Niepert, 2018], for the later we obtained this information from dumps found online on YAGO's website[3].

The FB15K-237 dataset contains a total of 29,395 numerical facts divided in 116 different numerical predicates. We evaluate our models on the top 10 numerical attributes ranked by the number of data samples. This reduces the dataset to 22,929 samples. We split these numerical facts into training, validation and test in the proportion of 80/10/10%, respectively. All other facts from FB15K-237 whose predicate belongs to $\mathcal{P}$ are used as training data, which amounts to 310,116 facts. Thus we only evaluate our approaches on their ability to answer queries whose answer is a numerical value.

The YAGO15K dataset contains 23,520 numerical facts divided in 7 different attributes. Similarly, we split these numerical facts into training, validation and test in the same proportion. We use all other 122,886 facts from this dataset for learning knowledge base embeddings. A summary of the datasets can be found in Table 1. All the splits of both datasets used in this work will be made publicly available to facilitate future comparisons.

---

3. https://www.mpi-inf.mpg.de/departments/databases-and-information-systems/research/yago-naga/yago/downloads/

| Dataset | Numerical Facts | | | Facts | $|\mathcal{E}|$ | $|\mathcal{P}|$ | $|\mathcal{A}|$ |
|---------|-------|-----|------|-------|------|-----|------|
| | train | dev | test | train | | | |
| FB15K-237 | 18,423 | 2,263 | 2,243 | 310,116 | 14,541 | 237 | 10 |
| YAGO15K | 18,872 | 2,330 | 2,318 | 122,886 | 15,404 | 32 | 7 |

Table 1: Dataset statistics.

We compare performance across methods using two evaluation metrics —MAE and RMSE. These are standard metrics used in regression problems and were also used in a related study of predicting numerics for language models [Spithourakis and Riedel, 2018].

$$err(e, a) = \mathbf{n}_e^a - \hat{\mathbf{n}}_e^a$$

$$MAE(a) = \frac{1}{|\mathcal{Q}^a|} \sum_{e \in \mathcal{Q}^a} |err(e, a)| \qquad RMSE(a) = \sqrt{\frac{1}{|\mathcal{Q}^a|} \sum_{e \in \mathcal{Q}^a} err(e, a)^2}$$

For TransE and TransE++ we fix the embedding dimension $d$ to 100. After some preliminary experiments, the weight $\alpha$ of TransE++ was fixed to 1. We used Adam [Kingma and Ba, 2014] to learn the parameters in a mini-batch setting with a learning rate of 0.001. We fixed the number of epochs to 100 and the mini-batch size to 256. The parameter $N$ of the negative sampling was set to 50. Within a batch, the number of data points for each of the TransE++'s regression objectives is proportional to the frequency of each of the numerical predicates in the training set. In all cases, the parameters were initialized following [Glorot and Bengio, 2010].

We used the Scikit-learn [Pedregosa et al., 2011] implementation of ridge regression for the approaches Lr and Lr++. The regularization term $\lambda_a$ is tuned using the values $[0, 0.1, 1, 10, 100]$.

For Nap and Nap++ the number of neighbors $(k)$ of the kNN graph is validated among $[3, 5, 10, 20]$; and the $\sigma$ of the RBF kernel is validated among $[0.25, 0.5, 1, 10]$.

All of the above is validated for each numerical predicate and evaluation metric.

## 5. Results

The objectives of this section is twofold: First we investigate the performance of our approaches (Lp and Nap, and their variants) with respect to the baselines. And second we experimentally check how robust these methods are for different degrees of sparsity in the training data.

Tables 2 and 3 detail the performance of the baselines and our approaches on FB15K-237. For each numerical attribute we always indicate in bold font the best performing method, which happens to be either Nap++ or Nap most of the time. Interestingly enough, from Table 2 we observe that for the numerical attributes 'location.area' and 'population.number' Global largely outperforms Local. This seems to indicate that the relational structure of this data set does not relate to these two numerical predicates. Overall, predictions for all other numerical attributes tend to benefit from the local information given by the entities' neighborhood. In comparing Tables 2 and 3, we note that Local is very competitive in

| | GLOBAL | | LOCAL | |
| --- | --- | --- | --- | --- |
| Num. Attribute | MAE | RMSE | MAE | RMSE |
| location.area | $\sim \mathbf{3.1e^4}$ | $\sim \mathbf{5.4e^5}$ | $\sim 3.7e^6$ | $\sim 6.9e^6$ |
| latitude | 9.99 | 16.67 | 3.38 | 10.30 |
| date_of_birth | 31.40 | 124.07 | 19.76 | 54.20 |
| population_number | $\sim \mathbf{3.9e^6}$ | $\sim \mathbf{1.7e^6}$ | $\sim 1.1e^7$ | $\sim 3.9e^7$ |
| person.height_mt | 0.085 | 0.104 | 0.091 | 0.113 |
| film_release_date | 12.29 | 16.71 | 10.75 | 15.54 |
| longitude | 52.10 | 68.28 | **5.32** | 16.32 |
| org.date_founded | 72.18 | 121.013 | 79.4 | 133.24 |
| date_of_death | 33.69 | 71.51 | 27.64 | 68.37 |
| location.date_founded | 120.66 | 259.84 | 136.23 | 552.84 |

Table 2: Performance of GLOBAL and LOCAL on FB15K-237.

| | LR | | NAP | | LR++ | | NAP++ | |
| --- | --- | --- | --- | --- | --- | --- | --- | --- |
| Num. Attribute | MAE | RMSE | MAE | RMSE | MAE | RMSE | MAE | RMSE |
| location.area | $\sim 7.7e^5$ | $\sim 1.0e^6$ | $\sim 5.1e^5$ | $\sim 1.5e^6$ | $\sim 8.9e^5$ | $\sim 1.2e^6$ | $\sim 2.9e^5$ | $\sim 8.5e^5$ |
| latitude | 8.47 | 12.44 | 2.5 | 5.83 | 6.52 | 10.85 | **2.20** | **5.02** |
| date_of_birth | 26.60 | 116.58 | 16.42 | 78.18 | 25.73 | 109.66 | **12.16** | **23.71** |
| population_number | $\sim 7.9e^6$ | $\sim 1.7e^7$ | $\sim 7.4e^6$ | $\sim 2.3e^7$ | $\sim 1.0e^7$ | $\sim 1.9e^7$ | $\sim 8.0e^6$ | $\sim 3.3e^7$ |
| person.height_mt | **0.065** | **0.083** | 0.074 | 0.092 | 0.073 | 0.091 | 0.074 | 0.092 |
| film_release_date | 5.59 | 7.68 | 4.13 | 6.35 | 5.78 | 7.68 | **3.90** | **5.81** |
| longitude | 25.56 | 34.69 | 6.22 | **16.04** | 24.77 | 33.29 | 6.26 | 21.26 |
| org.date_founded | 53.85 | 90.11 | **51.28** | **84.99** | 56.04 | 100.58 | 53.47 | 92.75 |
| date_of_death | 35.89 | 56.84 | 24.77 | 62.62 | 37.27 | 48.71 | **19.535** | **33.324** |
| location.date_founded | 145.46 | 227.09 | **79.65** | **161.02** | 139.29 | 240.26 | 88.04 | 201.88 |

Table 3: Performance of LR- and NAP-based models on FB15K-237.

regard to the numerical attributes 'latitude' and 'longitude'. This can be explained by the presence of predicates such as 'location.adjoins' or 'location.contains' in the relational structure of the graph. Similarly, entities' neighborhoods are useful for predicting 'date_of_birth' or 'date_of_death' because (some of the) surrounding entities correspond to people who have similar birth or death dates. Interestingly, all our approaches beat both baselines in the numerical attribute 'person.height_mt', for which *a priori* one would not expect performance gains in learning from the graph structure.

Overall, LR++ and NAP++ outperform their counterparts LR and NAP, respectively, for most numerical predicates. As we argued in Section 3.2.3 it is not feasible to validate the regularization term $\lambda_a$ for every numerical attribute while learning TRANSE++. We speculate that setting $\lambda_a = 0$ while training TRANSE++ may explain why LR++ and NAP++ do not always beat their counterparts.

| $P_r$ | 100 | | 80 | | 50 | | 20 | |
|---|---|---|---|---|---|---|---|---|
| **Num. Attribute** | Local | Nap++ | Local | Nap++ | Local | Nap++ | Local | Nap++ |
| location.area | $\sim 3.7\mathrm{e}^6$ | $\sim 2.9\mathrm{e}^5$ | $\sim 3.5\mathrm{e}^6$ | $\sim 5.0\mathrm{e}^5$ | $\sim 1.2\mathrm{e}^6$ | $\sim 4.0\mathrm{e}^5$ | $\sim 2.2\mathrm{e}^6$ | $\sim 2.3\mathrm{e}^5$ |
| latitude | 3.38 | **2.20** | 3.85 | **2.19** | 5.08 | **3.28** | 7.206 | **4.40** |
| date_of_birth | 19.76 | **12.16** | 23.63 | **15.444** | 22.96 | **12.93** | 27.2 | **19.20** |
| population_number | $\sim 1.1\mathrm{e}^7$ | $\sim 8.0\mathrm{e}^6$ | $\sim 1.2\mathrm{e}^7$ | $\sim 6.0\mathrm{e}^6$ | $\sim 9.09\mathrm{e}^6$ | $\sim 4.7\mathrm{e}^6$ | $\sim 5.2\mathrm{e}^7$ | $\sim 1.6\mathrm{e}^7$ |
| person.height_mt | 0.091 | **0.074** | 0.094 | **0.073** | 0.092 | **0.076** | 0.096 | **0.008** |
| film_release_date | 10.75 | **3.90** | 10.88 | **4.36** | 10.83 | **4.33** | 11.303 | **4.69** |
| longitude | **5.32** | 6.26 | 8.53 | **6.2** | 18.29 | **9.7** | 31.857 | **10.57** |
| org.date_founded | 79.4 | **53.47** | 74.68 | **50.3** | 81.77 | **52.59** | 73.35 | **61.44** |
| date_of_death | 27.64 | **19.54** | 27.46 | **21.74** | 25.33 | **23.42** | 35.06 | **29.42** |
| location.date_founded | 136.2 | **88.04** | 136.6 | **117.07** | 88.79 | **87.87** | 109.7 | **101.9** |

Table 4: Performance of Local and Nap++ on FB15K-237 for different degrees of sparsity, $P_r$, on the numerical facts. Results are reported in terms of Mean Absolute Error (MAE).

Another observation from Table 3 is that, in general, Nap-based models perform much better compared to Lr-based models. One can find a number of explanations to this. The obvious explanation is that the numerical attribute propagation approaches learn from labeled and unlabeled data, whereas the regression models only learn from labeled data. A second explanation is that whereas Nap's predictions are computed as a weighted average of observed numerical values, Lr's predictions are not bounded. This prevents Nap-based approaches from making large mistakes. On the other hand, for example, we observed non-plausible values (e.g. $> 2020$) predicted by the Lr-based models for the numerical attribute 'date_of_birth'. We also experimented with non-linear regression models, but did not observe any performance improvement.

Knowledge graphs are known to suffer from data sparsity due to missing facts. The same incompleteness is also true for numerical facts. Therefore it is crucial to study model performance under a sparse data regime. We generate data sparsity by artificially removing numerical facts from the training set while keeping the validation and test sets unchanged. We keep the underlying knowledge graph $\mathcal{G}$ unchanged because we aim to isolate the effect of numerical fact sparsity. In other words, only a number of numerical facts are removed from the training set. We retained a percentage $P_r$ of training numerical facts and ran Local and Nap++ with the same experimental set-up. We experimented with the following values of $P_r$: [100[4], 80, 50, 20]%. We detail the results of these experiments in Table 4. Note that the performance of Local degrades more rapidly compared to Nap++ as the sparsity increases. Even in high regimes of sparsity, Nap++'s performance is remarkably robust.

Table 5 lists results for Global and Local in YAGO15K. As for FB15K-237, Local outperforms Global for most of the numerical attributes. This reinforces our assumption that the numerical attributes explain, to some extent, relation structure between entities. Table 6 depicts the performance of Local, Nap and Nap++ under different degrees of sparsity in YAGO15K. In the light of these numbers, we can conclude that the Nap-based

---

4. This corresponds to the case of no artificial sparsity in numerical facts.

| | GLOBAL | | LOCAL | |
| --- | --- | --- | --- | --- |
| **Num. Attribute** | **MAE** | **RMSE** | **MAE** | **RMSE** |
| date_of_death | 37.99 | 89.47 | 39.70 | 92.38 |
| happenedOnDate | 38.55 | 67.33 | 38.55 | 67.33 |
| latitude | 12.51 | 21.50 | 3.04 | 9.04 |
| longitude | 53.07 | 63.195 | 11.38 | 24.08 |
| date_of_birth | 25.24 | 66.10 | 23.94 | 65.78 |
| createdOnDate | 89.32 | 155.83 | 132.20 | 197.18 |
| destroyedOnDate | 31.54 | 60.08 | 30.97 | 59.42 |

Table 5: Performance of GLOBAL and LOCAL on YAGO15K.

| $P_r$ | 100 | | | 80 | | | 50 | | | 20 | | |
| --- | --- | --- | --- | --- | --- | --- | --- | --- | --- | --- | --- | --- |
| **Num. Attribute** | **Local** | **Nap** | **Nap++** | **Local** | **Nap** | **Nap++** | **Local** | **Nap** | **Nap++** | **Local** | **Nap** | **Nap++** |
| date_of_death | 39.70 | 39.64 | **35.1** | 40.76 | 40.51 | **37.38** | 39.34 | 40.14 | **38.20** | 41.59 | 39.8 | **39.25** |
| happenedOnDate | 38.55 | 48.77 | **34.4** | 38.41 | 51.08 | **31.23** | 37.75 | 49.99 | **32.76** | 37.57 | 54.17 | **32.14** |
| latitude | 3.04 | 2.16 | **1.77** | 4.00 | **2.48** | 2.54 | 6.32 | 3.21 | **2.90** | 9.69 | **3.71** | 4.2 |
| longitude | 11.38 | **3.45** | 3.62 | 16.74 | 5.54 | **4.62** | 24.07 | 6.29 | **6.06** | 40.09 | 10.26 | **10.1** |
| date_of_birth | 23.94 | 18.32 | **16.91** | 24.16 | 18.69 | **17.49** | 24.39 | 18.79 | **18.07** | 25.70 | 20.12 | **18.82** |
| createdOnDate | 132.2 | 67.68 | **65.25** | 104.1 | 69.54 | **65.74** | 109.5 | **71.44** | 72.22 | 138.2 | 72.25 | **71.63** |
| destroyedOnDate | 30.97 | 25.61 | **21.63** | 31.45 | 28.58 | **25.98** | 30.67 | 24.42 | **21.17** | 31.4 | 27.80 | **26.5** |

Table 6: Performance of LOCAL, NAP and NAP++ on YAGO15K for different degrees of sparsity, $P_r$, on the numerical facts. Results are reported in terms of Mean Absolute Error.

models are more robust than LOCAL to data sparsity. NAP++ achieves the best performance for most of the numerical attributes and degrees of sparsity. It performs remarkably well for the numerical attribute 'happenedOnDate' in comparison to NAP. Across all values of $P_r$, on average, NAP++ improves NAP's performance by 20 points (in mean absolute value) for 'happenedOnDate'.

We recognize that reporting model performance in absolute values complicates comparison since numerical attributes lie on different ranges of values. To have a better picture of performance gains we report percentage error reduction between NAP++ and the best performing baseline. For numerical attribute $a$, the percentage error reduction in MAE is computed as follows

$$\triangle_{MAE} = \frac{\min(MAE_{\text{LOCAL}}(a), MAE_{\text{GLOBAL}}(a)) - MAE_{\text{NAP++}}(a)}{\min(MAE_{\text{LOCAL}}(a), MAE_{\text{GLOBAL}}(a))} \times 100.$$

One can compute the percentage error reduction in terms of RMSE in a similar manner.

This is shown in Table 7 for $P_r = 100$. We do not include 'location.area' and 'population.number', as previous experiments indicate that they do not relate to the graph structure of FB15K-237. Overall, NAP++ significantly outperforms baselines for almost all

numerical attributes in both FB15K-237 and YAGO15K data sets. These results demonstrate that the embeddings learned from the graph structure are useful predictors of entity numerical attributes.

| Num. Attribute | FB15K-237 | | YAGO15K | |
|---|---|---|---|---|
| | $\triangle_{MAE}$ | $\triangle_{RMSE}$ | $\triangle_{MAE}$ | $\triangle_{RMSE}$ |
| happenedOnDate | — | — | 18.53 | 31.89 |
| createdOnDate | — | — | 29.95 | 15.80 |
| DestroyedOnDate | — | — | 30.11 | 43.80 |
| date_of_birth | 38.43 | 56.26 | 29.37 | 10.65 |
| latitude | 35.05 | 51.28 | 41.86 | 45.45 |
| longitude | -17.61 | -30.29 | 12.43 | 48.4 |
| date_of_death | 29.33 | 51.26 | 7.61 | 14.0 |
| person.height_mt | 12.94 | 11.54 | — | — |
| film_release_date | 63.71 | 62.60 | — | — |
| org.date_founded | 25.93 | 23.36 | — | — |
| location.date_founded | 27.03 | 22.31 | — | — |

Table 7: Percentage error reduction between NAP++ and the best performing baseline for each numerical attribute in FB15K-237 and YAGO15K. The higher the value, the better the performance of NAP++ relative to the baselines.

| Entity $e$ | $\mathfrak{n}_e^a$ | $\hat{\mathfrak{n}}_e^a$ | | Nearest Neighbors | | | |
|---|---|---|---|---|---|---|---|
| | | Nap | Nap++ | Nap | | Nap++ | |
| Alexander the Great | -355 | 669 | **-108** | Aristotle (-384) | Kant (1724) | Julius Caesar (-100) | Aristotle (-384) |
| Galileo Galilei | 1564 | 816 | **1442** | Aristotle (-384) | Avicenna (980) | Avicenna (980) | Aquinas (1225) |
| John Locke | 1632 | 1450 | **1677** | Rousseau (1712) | Aristotle (-384) | Spinoza (1632) | Hume (1711) |
| Christoph. Columbus | 1506 | 1920 | **1506** | Lou Costello (1959) | Carravagio (1610) | Da Vinci (1519) | Michelangelo (1564) |
| Frederick the Great | 1786 | 1938 | **1915** | Hitler (1945) | Josip Tito (1980) | Brahms (1833) | Kant (1904) |

Table 8: Qualitative comparison between Nap and Nap++. The three first rows correspond to queries where the numerical attribute is 'date_of_birth', whereas for the two last queries it is 'date_of_death'. The actual value of labeled entities for the corresponding numerical attribute is shown in parenthesis.

## 5.1 Qualitative Analysis

This last experimental section aims to provide some insights on the benefit of adding numerical information during the representation learning stage.

Table 3 shows a noteworthy behavior of these methods with respect to the numerical attributes 'date_of_birth' and 'date_of_death'. While the performance of both approaches is comparable in terms of MAE, their RMSE largely differ. It is known that the mean absolute error is an evaluation metric more robust to outliers than the root mean squared error. We

set out to inspect these outliers to shed light on the usefulness of incorporating numerical information in the embeddings.

Nap-based models leverage these embeddings to build a similarity graph on which numerical information is propagated via Eq. (6). The resulting predictions are the result of multiplying the matrix $\mathbf{M}^a$ by the observed numerical values. This matrix[5] determines which observed entities' numerical values to pay attention to. These *attention* values are different for Nap and Nap++ as the graph similarity is constructed with different embeddings.

We qualitatively compare Nap and Nap++ based on a number of predictions computed in the test set. For each method we compare the two labeled entities they pay the most attention to. For the sake of simplicity we refer to these two entities as *nearest neighbors*. This is shown in Table 8. An interesting observation is that for NAP the two nearest neighbors are always entities topically similar to the entity in the query. On the other hand the nearest entities retrieved by NAP++ are more meaningful with respect to the queried numerical attribute. This is seen in the first query: (Alexander_the_Great[6], date_of_birth, ?). While Nap pays the most attention to topically similar entities, Nap++ puts a high attention on Julius Caesar,[7] which is more meaningful in regard to the date of birth.

Nap++ uses Euclidean distance between vectors to build the k-nearest neighbor graph. Table 8 that subsets of entities latent factors could be encoding different relational and numerical information. For instance, a few dimensions of the entity embeddings encode location information, while others encode population information and so on. To exploit this we learned Mahalanobis metrics for capturing different entity similarities. We did this while learning knowledge base embeddings by using an additional nearest neighbor loss. It did slightly improve the performance for few attributes, but overall it did not make significant distance. We suggest that future work should address this research direction in greater depth.

## 6. Conclusions

We introduce a novel problem, namely numerical attribute prediction in knowledge bases. Contrary to link prediction, the answer to this new query type is a numerical value, and not an element from a (small) closed vocabulary. Our premise to this problem is that the relational structure of a KB can be partially explained by the numerical attribute values associated with entities. This allows for leveraging KB embedding methods to learn representations that are useful predictors of numerical attributes. An extensive set of experiments validates our premise. Furthermore, we also show that learning KB representations enriched with numerical attribute information are helpful for addressing this task. Finally, we believe that this new problem introduced in the paper will spur interest and deeper investigation from the research community.

---

5. Note that it is non-negative and is row normalized.
6. For all practical purposes he is deemed a philosopher in FB15K-237.
7. Julius Caesar belongs to profession Politician in FB15K-237

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
