# OpenReview forum: "Learning Numerical Attributes in Knowledge Bases"
_AKBC.ws/2019/Conference — AKBC 2019_

### Official Review · AnonReviewer3 · 2019-01-03
**interesting paper with decent contribution, but doubts about practical viability**

**Rating:** 6
**Confidence:** 4

**Review:**

The paper presents a method to predict the values
of numerical properties for entities where these properties
are missing in the KB (e.g., population of cities or height of athletes).
To this end, the paper develops a suite of methods, using
regression and learned embeddings. Some of the techniques
resemble those used in state-of-the-art knowledge graph completion,
but there is novelty in adapting these techniques
to the case of numerical properties.
The paper includes a comprehensive experimental evaluation
of a variety of methods.

This is interesting work on an underexplored problem,
and it is carried out very neatly.

However, I am skeptical that it is practically viable.
The prediction errors are such that the predicted values
can hardly be entered into a high-quality knowledge base.
For example, for city population, the RMSE is in the order
of millions, and for person height the RMSE is above 0.05
(i.e., 5cm). So even on average, the predicted values are
way off. Moreover, average errors are not really the decisive
point for KB completion and curation.
Even if the RMSE were small, say 10K for cities, for some
cities the predictions could still be embarrassingly off.
So a knowledge engineer could not trust them and would
hardly consider them for completing the missing values.

Specific comment:
The embeddings of two entities may be close for different
reasons, potentially losing too much information.
For example, two cities may have close embeddings because
they are in the same geo-region (but could have very different
populations) or because they have similar characteristics
(e.g., both being huge metropolitan areas). Likewise, two
athletes could be close because of similar origin and
similar success in Olympic Games, or because they played
in the same teams.
It is not clear how the proposed methods can cope with
these confusing signals through embeddings or other techniques.

---

> ### Author Response · Authors · 2019-01-22
> **Rebuttal Review#3**
>
> Thanks for your helpful review.
>
> Let us address individually each of your concerns:
>
> 1. Prediction errors: We agree that, for the data sets discussed in the paper, the performance is not up to industrial standard and thus this technique may not be useful “as-is” in the industry. We believe that this paper is the first step which will hopefully spur more interest in this problem, and would eventually lead to industry-ready algorithms. However, this (current) limitation is not specific to this new problem, but common to many research problems (e.g. link prediction methods, for the moment, are far from meeting industrial requirements).
>
> 2. Embeddings: This is again an excellent point. Please also see rebuttal for Reviewer #2. Subsets of latent factors capture signals related to different relational and numerical information. A metric learning approach that “selects” a group of latent factors relevant to a certain numerical attribute would make sense. However, during our experiments, we learned Mahalanobis metrics by incorporating an additional loss during the learning of knowledge graph embeddings, and it did not improve the overall performance. We will discuss this in the paper. Nevertheless, we suggest that future work should address this research direction in greater depth.

---

### Official Review · AnonReviewer2 · 2019-01-07
**Nice paper on prediction of numeric attributes in KBs.**

**Rating:** 8
**Confidence:** 4

**Review:**

The paper reports on the prediction of numerical attributes in knowledge bases, a problem that has indeed
received too little attention. It lays out the problem rather clearly and defines datasets, a number of baselines
as well as a range of embedding-based models that I like because they include both simple pipeline models
(learn embeddings first, predict numerical attributes later) and models that include the numerical attributes
into embedding learning. I also appreciate that the paper makes an attempt to draw together ideas from different
research directions.

Overall, I like the paper. It's very solidly done and can serve as excellent base for further studies. Of course, I also have comments/criticisms.

First, the authors state that one of the contributions of the paper is to "introduce the problem of predicting the value
of entities’ numerical attributes in KB". This is unfortunately not true. There is relatively old work by Davidov and
Rappoport (ACL 2010) on learning numeric attributes from the web (albeit without a specific reference to KBs), and
a more recent study specifically aimed at attributes from KBs (Gupta et al. EMNLP 2015) which proposed and modelled exactly the same task, including defining freely available FreeBase-derived datasets.
More generally speaking, the authors seem to be very up to date regarding approaches that learn embeddings directly
from the KB, but not regarding approaches that use text-based embeddings. This is unfortunate, since the
model that we defined is closely related to the LR model defined in the current paper, but no direct comparison is
possible due to the differences in embeddings and dataset.

Second, I feel that not enough motivation is given for some of the models and their design decisions. For example,
the choice of linear regression seems rather questionable to me, since the assumptions of linear regression (normal distribution/homoscedasticity) are clearly violated by many KB attributes. If you predict, say, country populations, the
error for China and India is orders of magnitude higher then the error for other countries, and the predictions are dominated by the fit to these outliers. This not only concerns the models but also the evaluation, because
MAE/RMSE also only make sense when you assume that the attributes scale linearly -- Gupta et al. 2015 use this
as motivation for using logistic regression and a rank-based evaluation.
I realize that you comment on non-linear regression on p.12 and give a normalized evaluation on p.13:
I appreciate that, even though I think that it only addresses the problem to an extent.

Similarly, I like the label propagation idea (p. 7) but I lack the intuition why LP should work on (all) numeric attributes.
If, say, two countries border each other, I would expect their lat/long to be similar, but why should their (absolute) GDP be similar? What is lacking here is a somewhat more substantial discussion of the assumptions that this (and the other)
models make about the structure of the knowledge graph and the semantics of the attributes.

Smaller comment:
* Top of p.5, formalization: This looks like f is a general function, but I assume that one f is supposed to be learned
   for each attribute? Either it should be f_a, or f should have the signature E x A -> R.
   (p4: Why is N used as a separate type for numeric attributes if the function f is then supposed to map into reals anyway?)

---

> ### Author Response · Authors · 2019-01-22
> **Rebuttal Review#2**
>
> Thanks a lot for your feedback.
>
> Let us address individually each of your concerns:
>
> 1. Previous work: Thank you for bringing the work by Davidov and Rappoport (ACL 2010)  and Gupta et al. (EMNLP 2015) to our attention, we were not aware of this work. We will elaborate on these papers in the related work section. Regarding Gupta et al. (EMNLP 2015), the paper uses word2vec embeddings of named entities as inputs to a number of regression models. Similar to us, they aim to predict numerical attributes of knowledge base entities. Different to us, they leverage text information to do so. This is important, as in our problem we do not assume the existence of information other than the own graph structure. This relates to knowledge bases where entities’ names are unknown/anonymized  (e.g. medical knowledge bases). Nevertheless, this is very interesting and nicely complements our approach of learning from KG relationships as opposed to free text. Similarly, Davidov and Rappoport leverages text and/or class label information.
>
> 2. Linear Regression: As you correctly observed, different output variables have different distributions. Ideally, for attributes that do not permit a normal distribution assumption one would need to validate (i) the power transformation to be applied to the data, and (ii) the regression function. This comes at an exponential cost, as this validation should be jointly done for a number of regression models. To circumvent this issue we used a common normalization procedure for all attributes. We also trained with L1-loss (Robust regression), which is more robust (w.r.t. outliers) than OLS, for all attributes but the results were similar.
>
> 3. Label Propagation: Our assumption is that if two entities relate to the rest of the world in a similar manner, then they have similar numerical attributes. Nevertheless, your observation is an excellent point.
>
> Label Propagation uses Euclidean distance between vectors to build the k-nearest neighbor graph. We speculate that subsets of entities’ latent factors could be encoding different relational and numerical information. For instance, it is possible that a few dimensions of the entity embeddings encode location information, while others encode population information and so on. For this reason, we learned Mahalanobis metrics for capturing different entity similarities. We did this while learning knowledge base embeddings by using an additional nearest neighbor loss. It did slightly improve the performance for few attributes, but overall it did not improve the results. We will include this discussion in the paper. Nevertheless, we suggest that future work should address this research direction in greater depth.

---

### Official Review · AnonReviewer1 · 2019-01-08
**Solid work, convincing experiments and results**

**Rating:** 8
**Confidence:** 2

**Review:**

The paper presents innovative work towards learning numerical attributes in a KB, which the authors claim to be the first of its kind. The approach leverages KB embeddings to learn feature representations for predicting missing numerical attribute values. The assumption is that data points that are close to each other in a vector (embeddings) space have the same numerical attribute value. Evaluation of the approach is on a set of highest ranked 10 numerical attributes in a QA setting with questions that require a numerical answer.

The paper is well written and the approach is explained  in full detail.

My only concern is the application of the method across numerical attributes of (very) different granularity and context. The authors briefly mention the granularity aspect in section 3 and point to a normalization step. However, this discussion is very brief and leaves perhaps some further questions open on this.

---

> ### Author Response · Authors · 2019-01-22
> **Rebuttal Review#1**
>
> Thanks for the positive and encouraging comments.
>
> Granularity:  You have raised a very valid concern regarding the granularity of different numerical attributes. We agree on that the normalization step is really important to this problem, as one has to jointly train a number of regression models (with shared input vector representations) where output variables have different scales. We also tried min-max scaling, however this gave worse performance in comparison to standard scaling. We will add this comment to the paper.

---

### Meta-Review · Area_Chair1 · 2019-02-12
**New embedding method using numerical facts sparks interest but has some obvious limitations.**

**Recommendation:** Accept (Poster)
**Confidence:** 4

**Metareview:**

The authors consider the problem of predicting or imputing numerical attributes in knowledge bases. In contrast to simple local or global attribute prediction, they design a regression-based model that uses knowledge graph embeddings and extend the embedding computation to also use numerical attributes. Evaluation shows interesting preliminary results.

The critical consensus was that this paper is worthy of acceptance, but has several serious limitations that should be carefully explained. A critical issue is that the relationships used for propagating information do not necessarily correlate to similar numerical attribute values (e.g., geographical location is useful for predicting lat/long but not GDP or population), that a linear regression model is not sufficient to capture the full extent of relationships, that normalization of error values across different relationships skews the evaluation, several important and relevant related works were omitted, and the overall error rates are still somewhat high.

---

### Decision · Program_Chairs · 2019-02-15
**AKBC 2019 Conference Decision**

Accept